# Real-Time TDM-Guided Optimal Joint PK/PD Target Attainment of Continuous Infusion Piperacillin–Tazobactam Monotherapy Is an Effective Carbapenem-Sparing Strategy for Treating Non-Severe ESBL-Producing *Enterobacterales* Secondary Bloodstream Infections: Findings from a Prospective Pilot Study

**DOI:** 10.3390/microorganisms12010151

**Published:** 2024-01-12

**Authors:** Milo Gatti, Cecilia Bonazzetti, Renato Pascale, Maddalena Giannella, Pierluigi Viale, Federico Pea

**Affiliations:** 1Department of Medical and Surgical Sciences, Alma Mater Studiorum University of Bologna, 40138 Bologna, Italy; milo.gatti2@unibo.it (M.G.); cecilia.bonazzetti@unibo.it (C.B.); renato.pascale2@unibo.it (R.P.); maddalena.giannella@unibo.it (M.G.); pierluigi.viale@unibo.it (P.V.); 2Clinical Pharmacology Unit, Department for Integrated Infectious Risk Management, IRCCS Azienda Ospedaliero-Universitaria of Bologna, 40138 Bologna, Italy; 3Infectious Disease Unit, Department for Integrated Infectious Risk Management, IRCCS Azienda Ospedaliero-Universitaria of Bologna, 40138 Bologna, Italy

**Keywords:** ESBL-producing *Enterobacterales*, bloodstream infections, piperacillin–tazobactam, continuous infusion, optimal joint PK/PD target attainment, microbiological eradication, real-time TDM-guided expert clinical pharmacological advice program

## Abstract

(1) Objectives: To assess the impact of optimal joint pharmacokinetic/pharmacodynamic (PK/PD) target attainment of continuous infusion (CI) piperacillin–tazobactam monotherapy on the microbiological outcome of documented ESBL-producing *Enterobacterlaes* secondary bloodstream infections (BSIs). (2) Methods: Patients hospitalized in the period January 2022–October 2023, having a documented secondary BSI caused by ESBL-producing *Enterobacterales*, and being eligible for definitive targeted CI piperacillin–tazobactam monotherapy according to specific pre-defined inclusion criteria (i.e., absence of septic shock at onset; favorable clinical evolution in the first 48 h after starting treatment; low–intermediate risk primary infection source) were prospectively enrolled. A real-time therapeutic drug monitoring (TDM)-guided expert clinical pharmacological advice (ECPA) program was adopted for optimizing (PK/PD) target attainment of CI piperacillin–tazobactam monotherapy. Steady-state plasma concentrations (C_ss_) of both piperacillin and tazobactam were measured, and the free fractions (*f*) were calculated based on theoretical protein binding. The joint PK/PD target attainment was considered optimal whenever the piperacillin *f*C_ss_/MIC ratio was >4 and the tazobactam *f*C_ss_/target concentration (C_T_) ratio was >1 (quasi-optimal or suboptimal if only one or neither of the two thresholds were achieved, respectively). Univariate analysis was carried out for assessing variables potentially associated with failure in achieving the optimal joint PK/PD target of piperacillin–tazobactam and microbiological eradication. (3) Results: Overall, 35 patients (median age 79 years; male 51.4%) were prospectively included. Secondary BSIs resulted from urinary tract infections as a primary source in 77.2% of cases. The joint PK/PD target attainment was optimal in as many as 97.1% of patients (34/35). Microbiological eradication occurred in 91.4% of cases (32/35). Attaining the quasi-optimal/suboptimal joint PK/PD target of CI piperacillin–tazobactam showed a trend toward a higher risk of microbiological failure (33.3% vs. 0.0%; *p* = 0.08) (4) Conclusions: Real-time TDM-guided optimal joint PK/PD target attainment of CI piperacillin–tazobactam monotherapy may represent a valuable and effective carbapenem-sparing strategy when dealing with non-severe ESBL-producing *Enterobacterales* secondary BSIs.

## 1. Introduction

Infections caused by extended-spectrum beta-lactamase (ESBL)-producing *Enterobacterales* represent a global health concern. Several epidemiological studies have shown that ESBL-producing *Enterobacterales* resistant to third-generation cephalosporins may represent up to 35% and 18% of *Klebsiella pneumoniae* and *Escherichia coli* clinical isolates, respectively [1,2,3,4]. In a recent meta-analysis, infections caused by ESBL-producing *Enterobacterales* were associated with higher mortality rates compared with those caused by non-ESBL-producing *Enterobacterales* [5].

In the MERINO trial, treatment of bloodstream infections (BSIs) caused by ceftriaxone-resistant *Escherichia coli* or *Klebsiella pneumoniae* with piperacillin–tazobactam did not reach a non-inferior mortality rate compared with meropenem [6]. However, a large debate still exists nowadays about which therapeutic choices should be preferred in this scenario [7,8,9,10,11,12], as argued both in the European Society of Clinical Microbiology and Infectious Diseases (ESCMID) guidelines and in the Infectious Diseases Society of America (IDSA) guidance [13,14]. On the one hand, in the MERINO trial, findings of the inferiority of the piperacillin–tazobactam arm could have been affected by using an intermittent infusion dosing scheme of 4.5 g every 6 h over 30 min [6], which could have been suboptimal, also considering that in a post hoc analysis several clinical isolates were found to be resistant to piperacillin–tazobactam at broth microdilution testing [15]. On the other hand, the worrisome ever-growing increase in carbapenem-resistant Gram-negative infections promoted by the selective pressure deriving from extensive carbapenem use [16,17] may call into question the potential role that carbapenem-sparing strategies based on piperacillin–tazobactam could have in some non-severe clinical scenarios of ESBL-producing *Enterobacterales* infections [10]. In this latter regard, several well-designed observational studies have shown that no significant difference exists in terms of the mortality rate between piperacillin–tazobactam and carbapenems in the treatment of secondary BSIs caused by ESBL-producing *Enterobacterales* [18,19,20,21,22,23,24,25,26,27,28,29,30,31,32,33,34,35,36,37]. Additionally, the use of piperacillin–tazobactam compared with that of carbapenems was associated with a lower occurrence of colonization and/or infection caused by multidrug-resistant (MDR) or carbapenem-resistant Gram-negative pathogens. In a retrospective observational multicentric study including 151 patients with ESBL-producing BSIs, Ng et al. found that piperacillin–tazobactam was associated with a significantly lower acquisition of MDR bacterial infections compared with carbapenems (7.4% vs. 24.6%; *p* = 0.01) [20]. Similarly, in a retrospective observational multicentric study including 186 patients affected by ESBL-producing bacteremic urinary infections, Sharara et al. reported a trend toward a lower rate of 30-day colonization, with carbapenem-producing *Enterobacterales* among those receiving piperacillin–tazobactam compared with those receiving carbapenems (2% vs. 8%; *p* = 0.09) [30].

Some studies showed that when using beta-lactam/beta-lactamase inhibitor combinations (BL/BLIc), attaining optimal joint pharmacokinetic/pharmacodynamic (PK/PD) targets of both the BL and the BLI may be beneficial in terms of both maximizing the clinical/microbiological outcome and preventing resistance development [38,39]. In this scenario, implementing a real-time therapeutic drug monitoring (TDM)-based expert clinical pharmacological advice (ECPA) program may represent a valuable approach for assessing the optimal joint PK/PD target attainment of BL/BLIc administered by continuous infusion (CI) [40].

The aim of this study was to assess whether the optimal joint PK/PD target attainment of CI piperacillin–tazobactam monotherapy could represent a valuable carbapenem-sparing strategy in treating patients with documented BSIs caused by ESBL-producing *Escherichia coli* or *Klebsiella pneumoniae*.

## 2. Materials and Methods

### 2.1. Study Design and Inclusion Criteria

This prospective study was carried out between 1 January 2022 and 30 October 2023 in the medical wards, surgical wards, or intensive care units (ICUs) of the IRCCS Azienda Ospedaliero-Universitaria of Bologna, Italy. The study was conducted according to the guidelines of the Declaration of Helsinki and approved by the local ethical committee (No. EM 232–2022_308/2021/Oss/AOUBo on 16 March 2022 and No. 894/2021/Oss/AOUBo on 15 November 2021). Signed informed consent was collected from each included patient. Patients were enrolled if they met the following inclusion criteria: (a) having a documented piperacillin–tazobactam fully susceptible ESBL-producing *Enterobacterales* secondary BSI (phenotypically identified as being resistant to ceftriaxone and/or cefotaxime, as previously reported [6]); and (b) being eligible for definitive CI piperacillin–tazobactam targeted monotherapy according to the following pre-defined inclusion criteria: absence of septic shock at onset; favorable clinical evolution during the first 48 h after starting empirical treatment with piperacillin–tazobactam; low–intermediate risk primary infection source, namely, urinary tract infection (UTI), biliary/intrabdominal infection (IAI), or catheter-related (CR) BSI. Table 1 summarizes the inclusion criteria.

All of the patients received piperacillin–tazobactam therapy optimized by means of a real-time TDM-guided ECPA program and at least one follow-up blood culture a minimum of after 48 h from starting therapy for assessing microbiological outcome.

### 2.2. Data Collection

Demographic data (age, sex, weight, height, and body mass index (BMI)), clinical/laboratory data (admission ward, Charlson Comorbidity Index (CCI), immunocompetence status, baseline creatinine clearance (CLCr), need for intermittent hemodialysis (IHD) or continuous renal replacement therapy (CRRT), occurrence of augmented renal clearance (ARC), and status of source control), microbiological data (type/site of infection, and ESBL-producing clinical isolate with an MIC value for piperacillin–tazobactam), piperacillin–tazobactam treatment data (dosing regimen at baseline, average piperacillin and tazobactam steady-state concentrations (C_ss_) during treatment, number of TDM-based ECPAs, recommended dosing adjustments at first and at subsequent ECPAs, and treatment duration), and microbiological/clinical outcome data (microbiological eradication/failure, eventual resistance development, 30-day relapse, clinical cure, and 30-day mortality rate) were prospectively collected.

Immunocompetence status was defined as depressed whenever one or more of the following conditions existed: need for long-term use of corticosteroids and/or of biologic and/or antineoplastic agents; occurrence of solid or hematologic malignancies; previous solid organ (SOT) or hematopoietic stem cell transplantation (HSCT); and underlying HIV disease or autoimmune disease [41].

ARC was defined as a measured (based on 24 h urine collection) or an estimated (according to the CDK-EPI formula) creatinine clearance above 130 mL/min and 120 mL/min in males and females, respectively [42].

The status of source control was defined as failed whenever blood cultures were still positive seven days after the index culture, as previously reported [43].

The Centers for Disease Control and Prevention (CDC) criteria were adopted for defining the different types of infection [44]. Secondary BSIs were defined on the basis of [44] the simultaneous isolation of the same pathogen from at least one blood culture drawn by direct venipuncture and from the primary source, namely, from the peritoneal fluid, abdominal specimens, or bile in the case of IAI [45,46]; from urine cultures with a bacterial load of at least >10^5^ CFU/mL in the case of UTIs [44,45]; and from blood cultures drawn through a vascular device having 2 h shorter positivization time in the case of CR-BSI [44].

Piperacillin–tazobactam susceptibility was tested by means of a semi-automated broth microdilution method (Microscan Beckman NMDRM1). The European Committee on Antimicrobial Susceptibility Testing (EUCAST) clinical breakpoints were adopted for interpreting MIC results [47]. A threshold value of ≤8 mg/L identified *Enterobacterales* isolates susceptible to piperacillin–tazobactam [48].

### 2.3. Definition of Outcome Variables 

Microbiological outcomes were defined as eradication whenever the index pathogen was undetectable at the follow-up blood cultures, and, in contrast, as failure whenever the index pathogen was still detectable at the follow-up blood cultures (breakthrough BSI).

Resistance development was defined as an increase in the piperacillin–tazobactam MIC beyond the EUCAST clinical breakpoint of susceptibility.

Thirty-day relapse was defined as the re-growth of the index pathogen in blood cultures carried out within 30 days from stopping piperacillin–tazobactam therapy.

Clinical outcomes were defined as cures if a complete resolution of biochemical and clinical signs and symptoms of the infection was coupled with a documented microbiological eradication at the end of treatment and with an absence of relapse at the 30-day follow-up [49].

### 2.4. Piperacillin–Tazobactam Dosing Regimens, Sampling Procedure, and Implementation of a Real-Time TDM-Guided ECPA Program for Dosing Personalization

Piperacillin–tazobactam was started with a loading dose of 9 g administered over 2 h infusion immediately followed by a maintenance administered by CI [50] that was initially chosen according to the status of the patient’s renal function.

TDM of piperacillin and of tazobactam was assessed at steady-state (C_ss_), firstly after at least 24 h from starting therapy in order to be in steady-state conditions, and subsequently whenever feasible every 48–72 h during the whole treatment course. Total piperacillin and tazobactam C_ss_ were measured by means of a validated liquid chromatography–tandem mass spectrometry method [51]. By considering the plasma protein binding of piperacillin and of tazobactam reported in the literature, namely, 20% and 23%, respectively [52], the free (*f*) C_ss_ were calculated by multiplying the total C_ss_ by 0.80 and 0.77, respectively.

A real-time TDM-guided ECPA program supported by skilled MD Clinical Pharmacologists was used for optimizing the joint piperacillin–tazobactam PK/PD target in each individual patient, as previously reported [40].

### 2.5. Definition of Optimal, Quasi-Optimal, and Suboptimal Joint PK/PD Target Attainment of Piperacillin–Tazobactam

The PD determinant selected for assessing the efficacy of piperacillin–tazobactam monotherapy was a joint PK/PD target, as previously described [38]. Briefly, the joint PK/PD target was considered optimal whenever the piperacillin *f*C_ss_/MIC ratio was >4 and the tazobactam *f*C_ss_/target concentration (C_T_) ratio was >1 (where C_T_ is the fixed tazobactam target concentration of 4 mg/L proposed by the EUCAST for testing the in vitro susceptibility of the piperacillin–tazobactam combination); it was considered quasi-optimal whenever only one of the two thresholds was attained and suboptimal whenever none of the two was attained [38].

In patients having multiple TDM-guided ECPA programs during the treatment course, the average piperacillin and tazobactam *f*C_ss_ were calculated. The impact of the quality of the joint PK/PD target attainment of CI piperacillin–tazobactam on the microbiological outcome was then investigated.

### 2.6. Statistical Analysis

Continuous data were expressed as the median and interquartile range (IQR), whereas categorical variables were presented as counts or percentages. Univariate analyses (carried out by means of the Fisher’s exact test or the chi-squared test in cases of categorical variables, or by means of the Mann–Whitney U test in cases of continuous variables) were performed for comparing the potential correlation of the patients’ variables in attaining optimal vs. quasi-optimal/suboptimal piperacillin–tazobactam joint PK/PD targets, and in achieving microbiological eradication vs. microbiological failure. Statistical analyses were performed by means of MedCalc for Windows (MedCalc statistical software Ltd., version 19.6.1, Ostend, Belgium), and significance was defined as a *p* value < 0.05.

## 3. Results

Overall, a total of 35 hospitalized patients received definitive monotherapy with TDM-guided CI piperacillin–tazobactam for treating secondary ESBL-producing *Enterobacterales* BSI during the study period. Demographics and clinical features of the included patients are reported in Table 2.

The median (IQR) age was 79 years (68–85 years), with a slight male preponderance (51.4%). The median (IQR) CCI was 6 points (5–8.5 points), and 42.9% of cases were immunodepressed. Most patients were admitted to medical wards (21/35; 60.0%), and 10 (28.6%) were admitted to ICUs.

The median (IQR) baseline CLCr was 34 mL/min/1.73 m^2^ (19–47 mL/min/1.73 m^2^). Three patients (8.6%) underwent CRRT or IHD, and none experienced ARC.

The vast majority of secondary BSIs were related to UTIs (27/35; 77.2%); four were related to IAI, and four were related to CR-BSI (11.4%) as the primary source. Most patients (30/35; 85.7%) had effective source control. *Escherichia coli* and *Klebsiella pneumoniae* were the only two species of *Enterobacterales* isolated, and were detected in 22 and 13 of the index blood cultures, respectively. Most isolates exhibited an MIC value of 8 mg/L, namely, borderline with the EUCAST clinical breakpoint (25/35; 71.4%).

Piperacillin–tazobactam was administered at a median (IQR) daily dose of 9 g (6.75 g–13.5 g); the median (IQR) treatment duration was 10 days (7.25–13.75 days). Median (IQR) piperacillin and tazobactam *f*C_ss_ were 66.1 mg/L (37.1–99.0 mg/L) and 8.6 mg/L (5.4–14.9 mg/L), respectively. The median piperacillin *f*C_ss_/MIC ratio and the median tazobactam *f*C_ss_/C_T_ ratio were 8.9 (5.6–13.8) and 2.2 (1.4–3.7), respectively.

In total, 78 TDM-based ECPA programs for personalizing the CI piperacillin–tazobactam dosing regimen were performed, with a median (IQR) number of 2 (1–3) per patient. At first TDM-based ECPA, dosing reduction was recommended in the majority of cases (32/35; 91.4%). Overall, dosing adjustments were recommended in 46 out of 78 TDM-based ECPAs (59.0%), with three increases (3.9%) and 43 decreases (55.1%). Notably, optimal joint PK/PD target of piperacillin–tazobactam was attained in as many as 97.1% of cases (34/35), (in 1 case (2.9%) this was quasi-optimal; never suboptimal).

Microbiological eradication was obtained in 32 out of 35 cases (91.4%), whereas failure occurred in 3 cases (8.6%) (2 breakthrough BSIs and 1 30-day relapse). Developed resistance to piperacillin–tazobactam occurred only in one case (2.9%). Clinical cure was documented in 26 patients (74.3%), and the 30-day mortality rate was 8.6%.

Univariate analysis assessing variables potentially associated with microbiological eradication vs. failure is summarized in Table 3.

Only quasi-optimal/suboptimal joint PK/PD target attainment of piperacillin–tazobactam showed a trend toward a higher risk of microbiological failure compared with optimal joint PK/PD target attainment (33.3% vs. 0.0%; *p* = 0.08; Figure 1).

## 4. Discussion

To the best of our knowledge, this is the first prospective study that has explored the relationship between a joint PK/PD target attainment of CI piperacillin–tazobactam and the microbiological outcome among hospitalized patients receiving definitive CI piperacillin–tazobactam monotherapy for treating non-severe ESBL-producing *Enterobacterales* secondary BSIs. Notably, the findings showed that in the vast majority of patients, real-time TDM-guided ECPA programs of CI piperacillin–tazobactam facilitated optimal joint PK/PD target attainment and microbiological eradication.

In a recent retrospective study carried out among 43 ICU critically ill patients having documented Gram-negative BSI and/or ventilator-associated pneumonia, we showed that the TDM-guided attainment of optimal joint PK/PD target of CI piperacillin–tazobactam monotherapy granted very high microbiological eradication rates (87.4%) and resulted in protection against microbiological failure (OR 0.03; 95%CI 0.003–0.27; *p* = 0.002) [53]. Although in that study the number of patients having ESBL-producing *Enterobacterales* infections was quite limited (only 6/43), the findings allowed us to hypothesize that this strategy could have been potentially effective even when dealing with ESBL producers [53].

Indeed, using definitive piperacillin–tazobactam monotherapy for treating ESBL-producing *Enterobacterales* BSIs was called into question by the findings of the MERINO trial, showing that, in this setting, piperacillin–tazobactam use was associated with higher mortality rates compared with meropenem use [6]. However, arguments that the results of the MERINO trial could have been at least partially affected by the fact that PK/PD target attainment of piperacillin–tazobactam could have been suboptimal due to intermittent infusion administration should not be overlooked [10].

Consequently, in the post-MERINO trial era, some guidance and/or viewpoints started suggesting that piperacillin–tazobactam administered by extended-infusion or, even better, by CI, could have represented a valuable option for treating ESBL-producing *Enterobacterales* secondary BSIs, especially whenever originating from sources at low–intermediate infection risk, namely, UTIs, IAIs, or CR-BSIs, being non-severe, and being caused by fully susceptible piperacillin–tazobactam strains [10,11,13,54].

The findings of our study may support the contention that administering piperacillin–tazobactam by CI and optimizing the joint PK/PD target attainment in real time thanks to a TDM-guided ECPA program may result in very high microbiological eradication rates among patients affected by non-severe secondary BSIs, even when caused by ESBL-producing *Enterobacterales*. Notably, in our study, we have introduced the innovative concept of joint PK/PD target for optimizing piperacillin–tazobactam therapy. According to this, in order to maximize clinical efficacy and prevent microbiological failure, it is important to attain an optimal PK/PD target not only for piperacillin, namely, the beta-lactam, but also for tazobactam, namely, the beta-lactamase inhibitor. In this regard, administering piperacillin–tazobactam by CI and adapting dosing regimens based on a TDM-guided approach may both maximize the PK/PD target of piperacillin to 100%*f*T_>4–8×MIC_ and steadily maintain the tazobactam C_ss_ above the fixed tazobactam target concentration of 4 mg/L proposed by the EUCAST for testing the in vitro susceptibility of the piperacillin–tazobactam combination [55]. In the scenario of challenging clinical conditions, such as those of BSIs due to ESBL-producing *Enterobacterales*, attaining an optimal PK/PD target for both piperacillin and tazobactam could represent a major driver for improving both the clinical efficacy and prevention of Gram-negative resistance occurrence as much as possible, considering that different preclinical studies have reported a consistent decrease in piperacillin MIC values in the presence of a tazobactam concentration increase [56,57,58]. Particularly, in a hollow-fiber infection model in which different ESBL-producing clinical isolates were tested, the attainment of a piperacillin–tazobactam exposure of %*f*T > instantaneous MIC (MIC_i_; namely, the changing pathogen susceptibility in the presence of changing inhibitor concentrations) higher than 55.1–73.6% was significantly associated with the prevention of bacterial regrowth [56,57,58]. Indeed, the confirmatory findings of this being a very suitable subset of patients in which applying this approach could represent a valuable carbapenem-sparing option, enabling improved antimicrobial stewardship programs focused at decreasing carbapenem use in settings with a high prevalence of carbapenemase-producing Gram-negatives [10]. Obviously, appropriate source control should be mandatory in this context for minimizing either the risk of microbiological failure or that of relapse occurrence, as previously reported [54,59].

Finally, it should also be mentioned that the availability of a real-time TDM-guided ECPA program may prove to be extremely helpful in promptly recognizing and correcting cases having only quasi-optimal/suboptimal joint PK/PD target attainment of CI piperacillin–tazobactam. Indeed, the implementation of a real-time TDM-guided strategy was significantly associated with higher target attainment rates compared with the standard approach [60].

Limitations of our study should be acknowledged. The study design was monocentric, and the sample size was quite limited. Conversely, the prospective design is a point of strength, as was enrolling patients receiving piperacillin–tazobactam monotherapy. This confirmed the valuable role of piperacillin–tazobactam as a carbapenem-sparing strategy in this setting by avoiding any confounding bias on clinical and microbiological outcomes associated with the eventual use of combination therapy with other anti-Gram-negative antibiotics.

## 5. Conclusions

The preliminary findings of this prospective study suggest that real-time TDM-guided optimal joint PK/PD target attainment of CI piperacillin–tazobactam monotherapy may represent a valuable and effective carbapenem-sparing strategy when dealing with non-severe ESBL-producing *Enterobacterales* secondary BSIs. Larger definitive confirmatory studies are warranted.

## Figures and Tables

**Figure 1 microorganisms-12-00151-f001:**
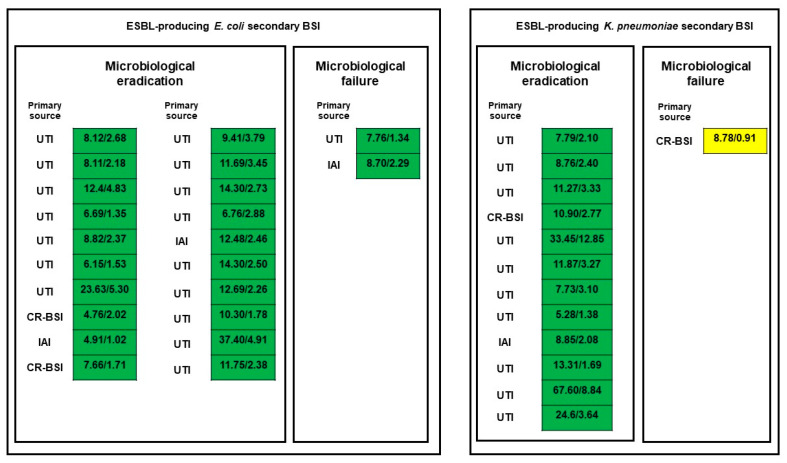
Relationship between microbiological outcome and optimal (green box), quasi-optimal (yellow box) or suboptimal (red box) joint PK/PD target attainment of piperacillin–tazobactam. A stronger trend toward higher microbiological failure rates was found among patients attaining the quasi-optimal/suboptimal joint PK/PD target of piperacillin–tazobactam than in those attaining the optimal target (33.3% vs. 0.0%; *p* = 0.08). CR-BSI: catheter-related bloodstream infection; IAI: intrabdominal infection; UTI: urinary tract infection.

**Table 1 microorganisms-12-00151-t001:** Summary of the inclusion criteria for prospectively treating monotherapy patients having secondary bloodstream infections caused by ESBL-producing *Enterobacterales* with piperacillin–tazobactam.

Variables	Inclusion Criteria
Pathogen	ESBL-producing *Enterobacterales* fully susceptible to piperacillin–tazobactam according to the EUCAST (i.e., MIC values ≤ 8 mg/L)
Antibiotic treatment	Piperacillin–tazobactam monotherapy by CI No additional agent with activity against Gram-negative pathogens was allowed (namely, aminoglycosides, fluoroquinolones, fosfomycin, tigecycline, and/or colistin)
Type of infection	Secondary BSI originating from sources at low–intermediate infection risk (namely, UTIs, IAIs, or CR-BSIs-BSI), and effective source control
Severity of infection at presentation	Non-severe infections without septic shock at onset occurring both in non-ICU-admitted patients and in ICU-admitted patients Favorable clinical response within the first 48 h after starting empirical treatment with piperacillin–tazobactam

BSI: bloodstream infection; CI: continuous infusion; CR-BSI: catheter-related bloodstream infection; ESBL: extended spectrum beta-lactamase; EUCAST: European Committee on Antimicrobial Susceptibility Testing; IAI: intrabdominal infection; ICU: intensive care unit; MIC: minimum inhibitory concentration; UTI: urinary tract infection.

**Table 2 microorganisms-12-00151-t002:** Demographic, clinical characteristics, and piperacillin–tazobactam treatment features of the included patients having definitive TDM-guided CI piperacillin–tazobactam monotherapy for treating BSIs caused by ESBL-producing *Enterobacterales*.

Demographics and Clinical Variables	Patients (N = 35)
Patient demographics	
Age (years) (median (IQR))	79 (68–85)
Gender (male/female) (n (%))	18/17 (51.4/48.6)
Body weight (Kg) (median (IQR))	70 (62–75)
Body mass index (Kg/m^2^) (median (IQR))	24.2 (21.7–26.2)
Admission ward (n (%))	
Medical	21 (60.0)
Surgical	4 (11.4)
ICU	10 (28.6)
Underlying conditions	
Charlson Comorbidity Index (median (IQR))	6 (5–8.5)
Immunosuppression (n (%))	15 (42.9)
Status of renal function	
Baseline CL_CR_ (mL/min/1.73 m^2^) (median (IQR))	34 (19–47)
IHD/CRRT (n (%))	3 (8.6)
Augmented renal clearance (n (%))	0 (0.0)
Source of BSI (n (%))	
UTI	27 (77.2)
IAI	4 (11.4)
CR-BSI	4 (11.4)
Failure in achieving effective source control (n (%))	5 (14.3)
ESBL-producing Enterobacterales (n (%))	
*Escherichia coli*	22 (62.9)
*Klebsiella pneumoniae*	13 (37.1)
MIC value (n (%))	
4 mg/L	10 (28.6)
8 mg/L	25 (71.4)
Piperacillin–tazobactam treatment	
Daily dose (mg) (median (IQR))	9 g/day (6.75 g/day–13.5 g/day)
Treatment duration (days) (median (IQR))	10 (7.25–13.75)
Piperacillin *f*C_ss_ (mg/L) (median (IQR))	66.1 (37.1–99.0)
Tazobactam *f*C_ss_ (mg/L) (median (IQR))	8.6 (5.4–14.9)
Piperacillin *f*C_ss_/MIC ratio (median (IQR))	8.9 (5.6–13.8)
Tazobactam *f*C_ss_/C_T_ ratio (median (IQR))	2.2 (1.4–3.7)
PK/PD target attainment	
Overall optimal joint PK/PD target (n (%))	34 (97.1)
Overall quasi-optimal joint PK/PD target (n (%))	1 (2.9)
Overall suboptimal joint PK/PD target (n (%))	0 (0.0)
Overall optimal joint PK/PD target at first TDM assessment (n (%))	34 (97.1)
Overall quasi-optimal joint PK/PD target at first TDM assessment (n (%))	1 (2.9)
Overall suboptimal joint PK/PD target at first TDM assessment (n (%))	0 (0.0)
ECPA program	
Overall TDM-based ECPAs	78
Number of TDM-based ECPA programs per treatment course (median (IQR))	2 (1–3)
Number of dosage confirmations at first TDM assessment (n (%))	3 (8.6)
Number of dosage decreases at first TDM assessment (n (%))	32 (91.4)
Number of dosage increases at first TDM assessment (n (%))	0 (0.0)
Overall number of dosage confirmations (n (%))	32 (41.0)
Overall number of dosage decreases (n (%))	43 (55.1)
Overall number of dosage increases (n (%))	3 (3.9)
Outcome	
Microbiological eradication (n (%))	32 (91.4)
Resistance development (n (%))	1 (2.9)
30-day relapse (n (%))	1 (2.9)
Clinical cure (n (%))	26 (74.3)
30-day mortality (n (%))	3 (8.6)

BSI: bloodstream infection; CL_CR_: creatinine clearance; CR-BSI: catheter-related bloodstream infection; CRRT: continuous renal replacement therapy; ECPA: expert clinical pharmacological advice; *f*C_ss_: free steady-state concentrations; *f*C_T_: free target concentrations; IAI: intrabdominal infection; ICU: intensive care unit; IHD: intermittent hemodialysis; IQR: interquartile range; MIC: minimum inhibitory concentration; PK/PD: pharmacokinetic/pharmacodynamic; TDM: therapeutic drug monitoring; UTI: urinary tract infection.

**Table 3 microorganisms-12-00151-t003:** Univariate analysis comparing patients’ variables potentially associated with microbiological eradication vs. microbiological failure.

Variables	Microbiological Eradication (N = 32)	Microbiological Failure (N = 3)	Univariate Analysis *p* Value
Patient demographics			
Age (years) (median (IQR))	79 (67.75–85)	78 (78–84)	0.58
Gender (male/female) (n (%))	16/16 (50.0/50.0)	2/1 (66.7/33.3)	0.99
Body weight (Kg) (median (IQR))	70 (62–75)	63 (61.5–71.5)	0.74
Body mass index (Kg/m^2^) (median (IQR))	24.2 (21.8–26.2)	24.6 (22.7–27.9)	0.81
Admission ward (n (%))			
Medical	20 (62.5)	1 (33.3)	0.55
Surgical	4 (12.5)	0 (0.0)	0.99
ICU	8 (25.0)	2 (66.7)	0.19
Underlying conditions			
Charlson Comorbidity Index (median (IQR))	6 (5–8.25)	6 (5.5–8.5)	0.81
Immunosuppression (n (%))	14 (43.8)	1 (33.3)	0.99
IHD/CRRT (n (%))	3 (9.4)	0 (0.0)	0.99
Source of BSI (n (%))			
UTI	26 (81.2)	1 (33.3)	0.12
IAI	3 (9.4)	1 (33.3)	0.31
CR-BSI	3 (9.4)	1 (33.3)	0.31
Failure in achieving complete source control (n (%))	4 (12.5)	1 (33.3)	0.38
ESBL-producing Enterobacterales (n (%))			
*Escherichia coli*	20 (62.5)	2 (66.7)	0.99
*Klebsiella pneumonia*	12 (37.5)	1 (33.3)	0.99
MIC value (n (%))			
4 mg/L	8 (25.0)	2 (66.7)	0.19
8 mg/L	24 (75.0)	1 (33.3)	0.19
Piperacillin–tazobactam treatment and joint PK/PD target attainment			
Quasi-optimal/suboptimal joint PK/PD target attainment	0 (0.0)	1 (33.3)	0.08

BSI: bloodstream infection; CR-BSI: catheter-related bloodstream infection; CRRT: continuous renal replacement therapy; IAI: intrabdominal infection; ICU: intensive care unit; IHD: intermittent hemodialysis; IQR: interquartile range; MIC: minimum inhibitory concentration; PK/PD: pharmacokinetic/pharmacodynamic; UTI: urinary tract infection.

## Data Availability

The data presented in this study are available on request from the corresponding author. The data are not publicly available due to privacy concerns.

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
