# Peer review of "Real-Time TDM-Guided Optimal Joint PK/PD Target Attainment of Continuous Infusion Piperacillin–Tazobactam Monotherapy Is an Effective Carbapenem-Sparing Strategy for Treating Non-Severe ESBL-Producing Enterobacterales Secondary Bloodstream Infections: Findings from a Prospective Pilot Study"

_microorganisms, 2024, doi:10.3390/microorganisms12010151_

Round 1

Reviewer 1 Report

Comments and Suggestions for Authors

Is 30 days of treatment feasible? Should it be different depending on the patient's condition?

Does the use of other medications during treatment affect the outcome of bacterial retention? Appropriate comments or comments are suggested.

Is there more to be done to evaluate the outcome of the cure? Or provide more test data.

Author Response

Manuscript ID: microorganisms-2814507 entitled “Real-time TDM-guided optimal joint PK/PD target attainment of continuous infusion piperacillin-tazobactam monotherapy is an effective carbapenem-sparing strategy for treating non-severe ESBL-producing Enterobacterales secondary bloodstream infections: findings from a pilot prospective study” by Gatti et al.

Dear Editor,

We would like to thank you for the opportunity to resubmit a revised version of this manuscript. We appreciated the reviewers’ constructive comments. All have been carefully considered and addressed in the revision, where and whenever possible. Furthermore, as per request by the Editorial Office, we extended the main text over 4000 words and reduced self-citation rate below 15%.

Our point-by-point responses are provided below.

Q= QUERY; A= ANSWER

Reviewer #1

Q1. Is 30 days of treatment feasible? Should it be different depending on the patient's condition?

A1. We think that a misinterpretation could have occurred, since as reported in Table 2 (previously Table 1) the median piperacillin-tazobactam treatment duration was of only 10 days (IQR 7.25-13.75 days). The 30-day threshold was referred to the time for assessing eventual microbiological relapse, as just reported previously in other studies (refer to 10.1001/jama.2020.6348; 10.3390/antibiotics9020071).

Q2. Does the use of other medications during treatment affect the outcome of bacterial retention? Appropriate comments or comments are suggested.

A2. We thank the reviewer for this comment, allowing us to better clarify this important issue. As reported in the Methods section (refer to Line 105), piperacillin-tazobactam monotherapy was a predefined enrollment criteria of this prospective study, so that no other antimicrobial treatment could have affected the microbiological outcome. This aspect was reported in the Discussion section as a point of  strength of our study (refer to Line 341-344).

Q3. Is there more to be done to evaluate the outcome of the cure? Or provide more test data.

A3. We respectfully highlight that the type and the numerousness of the indicators of clinical outcomes included for assessing the efficacy of piperacillin-tazobactam monotherapy in the treatment of bloodstream infections caused by ESBL-producing Enterobacterales is in agreement with what just previously reported in previous studies (refer to doi.org/10.1093/cid/ciab176; 10.3390/antibiotics9020071) and should therefore be considered appropriate. The included indicators were microbiological eradication, resistance development to piperacillin-tazobactam, 30-day relapse, clinical cure and 30-day mortality.

Reviewer 2 Report

Comments and Suggestions for Authors

Milo Gatti  and collaborators reported an original one center prospective study included 35 patients with TDM guided CI piperacillin- tazobactam for treating non-severe secondary ESBL producing Enterobacteriales BSI . The manuscript is informative and  presents still very important and interesting issue in the area of infection treatment with TDM and PK/PD dose modification usage. Despite the numerous articles in this area  had been published, and from the probably first in real time CI PIP/TAZ with dose modification pilot study (don’t cited in this manuscript published by Duszynska W, FS Taccone  and others  in Int. J. Antimicrob. Agents 2012, 39(2)153-8 ))  a lot of time has passed, the topic of this study is still up to date . The most original aspect of this manuscript is still interesting methodology ( PIP/TAZ in CI with dose modification in real time on a basis ECPA program  and the impact on microbiological outcome (eradication) , resistance development, clinical cure, 30-day mortality rate. Nevertheless , infections caused by ESBL producing Enterobacteriales are usually (at severe ill ICU patients) treated by carbapenems , ceftazidime /avibactame, aminoglycosides while PIP/Taz is use currently rare ( instead of  UTI ) because a global resistance of G(-) pathogens.

Overall, the manuscript adheres to relevant standards for reporting and data deposition .I congratulate the authors ideas of this study .I can recommend this manuscript for publication after this small correction.

I have only minor comments  :

 Table 1 Please change a little table description because in the table are also  pip/taz treatment characteristic and results of treatment.

Page 3 line 146  add please “ biochemical and clinical “ symptoms …

Add please , when CI was started .Immediately after LD ?…

Add please, how frequently and how long Css was measured ? During all the treatment period?

Author Response

Manuscript ID: microorganisms-2814507 entitled “Real-time TDM-guided optimal joint PK/PD target attainment of continuous infusion piperacillin-tazobactam monotherapy is an effective carbapenem-sparing strategy for treating non-severe ESBL-producing Enterobacterales secondary bloodstream infections: findings from a pilot prospective study” by Gatti et al.

Dear Editor,

We would like to thank you for the opportunity to resubmit a revised version of this manuscript. We appreciated the reviewers’ constructive comments. All have been carefully considered and addressed in the revision, where and whenever possible. Furthermore, as per request by the Editorial Office, we extended the main text over 4000 words and reduced self-citation rate below 15%.

Our point-by-point responses are provided below.

Q= QUERY; A= ANSWER

Reviewer #2

Milo Gatti  and collaborators reported an original one center prospective study included 35 patients with TDM guided CI piperacillin- tazobactam for treating non-severe secondary ESBL producing Enterobacteriales BSI. The manuscript is informative and presents still very important and interesting issue in the area of infection treatment with TDM and PK/PD dose modification usage. Despite the numerous articles in this area had been published, and from the probably first in real time CI PIP/TAZ with dose modification pilot study (don’t cited in this manuscript published by Duszynska W, FS Taccone and others in Int. J. Antimicrob. Agents 2012, 39(2)153-8 )) a lot of time has passed, the topic of this study is still up to date. The most original aspect of this manuscript is still interesting methodology (PIP/TAZ in CI with dose modification in real time on a basis ECPA program and the impact on microbiological outcome (eradication) , resistance development, clinical cure, 30-day mortality rate. Nevertheless , infections caused by ESBL producing Enterobacteriales are usually (at severe ill ICU patients) treated by carbapenems, ceftazidime /avibactame, aminoglycosides while PIP/Taz is use currently rare ( instead of  UTI ) because a global resistance of G(-) pathogens.

Overall, the manuscript adheres to relevant standards for reporting and data deposition .I congratulate the authors ideas of this study .I can recommend this manuscript for publication after this small correction.

We thank the reviewer for appreciating our study.

I have only minor comments  :

Q1. Table 1 Please change a little table description because in the table are also  pip/taz treatment characteristic and results of treatment.

A1. Thank you for this suggestion. We modified the description of Table 2 (previously Table 1) accordingly.

Q2. Page 3 line 146  add please “ biochemical and clinical “ symptoms …

A2. Thank you for this suggestion. We modified accordingly.

Q3. Add please , when CI was started .Immediately after LD ?…

A3. We thank the reviewer for this comment, allowing us to better clarify this issue. CI was started immediately after loading (refer to the Methods section, Line 174).

Q4. Add please, how frequently and how long Css was measured ? During all the treatment period?

A4. We thank the reviewer for this comment. Piperacillin-tazobactam concentrations were measured firstly after at least 24 hours from starting therapy in order to be in steady-state conditions, as specified in the Methods section (refer to Line 177-179), and subsequently reassessed every 48-72 hours whenever feasible during the whole treatment period. We specified better these aspects in the Methods section (refer to Line 177-179).